# Exploring Pregnancy Outcomes Associated with SARS-CoV-2 Infection

**DOI:** 10.3390/medicina57080796

**Published:** 2021-08-01

**Authors:** Madalina Timircan, Felix Bratosin, Iulia Vidican, Oana Suciu, Livius Tirnea, Valentina Avram, Iosif Marincu

**Affiliations:** 1Department of Gynecology, “Victor Babes” University of Medicine and Pharmacy, 300041 Timisoara, Romania; timircan.madalina@yahoo.com; 2Methodological and Infectious Diseases Research Center, Department of Infectious Diseases, “Victor Babes” University of Medicine and Pharmacy, 300041 Timisoara, Romania; iulia.georgianabogdan@gmail.com (I.V.); oana_suciu96@yahoo.com (O.S.); liviustirnea@yahoo.com (L.T.); avramvalentinaelena@gmail.com (V.A.); imarincu@umft.ro (I.M.)

**Keywords:** adverse birth outcomes, COVID-19, pregnancy outcomes, SARS-CoV-2, preterm births

## Abstract

*Background and Objectives*: The ongoing pandemic proved to be a tremendous challenge to all economic layers, healthcare, and people safety. As more than one year elapsed since the beginning of the COVID-19 pandemic, a multitude of medical studies involving the SARS-CoV-2 virus helped researchers and medical practitioners in understanding the effects it has on all sorts of patients until effective vaccines were finally developed and distributed for mass vaccination. Still, the SARS-CoV-2 and its new variants remain a potential threat towards all categories of patients, including a more delicate group represented by pregnant women. Thus, the current study aims to investigate the potential effects on obstetrical outcomes after a positive SARS-CoV-2 infection. *Materials and Methods*: This single-center prospective cohort study investigated the pregnancy outcomes in a total of 1039 eligible pregnant women between 30 August 2020 and 30 January 2021. Multiple patient characteristics and obstetrical outcomes were tested and analyzed in a multivariate regression model to establish potential risks determined by a COVID-19-positive pregnancy towards the mother and the newborn. *Results*: In the study sample, there were 938 pregnancies included without COVID-19 and 101 pregnant women identified with a positive COVID-19 infection. COVID-19 was significantly associated with a 2-fold increase in the risk of premature rupture of membranes and 1.5 times higher risk of preterm birth with emergency c-sections and lower APGAR scores. Also, significantly more newborns were given birth prematurely, with lower APGAR scores after the mothers were infected with SARS-CoV-2. *Conclusions*: A third-trimester infection with SARS-CoV-2 is a significant risk factor for preterm birth via an emergency cesarean section, a premature rupture of membranes, and a lower APGAR score in newborns, as compared with pregnancies where COVID-19 was not identified.

## 1. Introduction

Coronavirus disease (COVID-19) caused by the severe acute respiratory syndrome coronavirus 2 (SARS-CoV-2) triggered a pandemic in 2020, making it responsible for an estimated 979 thousand excess deaths in 29 high-income countries [1], including the US and the UK, and a total of at least 3 million deaths worldwide by the end of 2020 [2]. As the ongoing pandemic progresses, COVID-19 seems to not only affect the respiratory tract but proves to be a multi-systemic disease, with a vast range of unspecific signs and symptoms [3]. A major concern during the COVID-19 pandemic was the risk of the infection itself and the risk of other diseases not being kept under careful observation or discovered promptly, since restrictions have decreased access to primary and specialty care [4]. Individuals with chronic diseases received less in-person care due to government regulations on planned and non-urgent medical consultations and increased worry of COVID-19 exposure during in-person appointments. Similarly, another sensitive group of patients requiring strict medical attention and planned follow-up comprised pregnant women. 

The COVID-19 pandemic caused global supply chain disruptions followed by significant shortages of medicines required to manage acute and chronic diseases [5]. Patients, including pregnant women, experienced disturbances in their lifestyles, most notably in physical activity, sleep, stress, and mental health, which need to be handled more effectively [6]. Considering the drop in admissions and the association between deferred care and increased morbidity, it is expected that future patients who postpone hospitalization will arrive with more severe conditions than usual [7]. Based on these observations, we hypothesize that pregnant women left behind for antenatal observation and care due to the restrictions imposed during the COVID-19 pandemic will have adverse pregnancy outcomes [8], such as higher numbers of preterm deliveries and other major neonatal morbidities. In 2015, the average percentage of preterm live births was around 8% across Europe, while Romania registered 8.4% of the average percentage of preterm live births [9]. The body of pregnant women suffers from immunological changes associated with birth, resulting in the susceptibility to and severity of certain infectious diseases. At the same time, the risk of preterm birth can be increased by acute or chronic infections, with around 50% of preterm deliveries occurring due to an inflammatory process [10]. It seems that the maternal transmission of SARS-CoV-2 is improbable, but there appears to be an underlying danger of placental insufficiency due to COVID-19 infection’s prothrombotic propensity [11]. 

Different studies have already investigated the impact of the pandemic on pregnancy outcomes, but there is still a lot of uncertainty regarding this topic since the findings differ broadly. What has been clear so far is that hospital admissions have decreased; for example, a study taking place during an interval of 10 weeks found that there was a 43.2 percent decrease in hospital admissions and a 66.4 percent decrease in reported obstetric emergencies compared to the previous year’s similar calendar months [12]. However, researchers found that either the proportions of preterm live births decreased during the pandemic [13] or that they did not differ during the lockdown [14], or the opposite, with higher preterm birth rates happening during the COVID-19 pandemic, compared with a period before the pandemic [15]. Thus, what they all have in common is analyzing a short period of two or three months during the full lockdown in the Spring of 2020 and a possible explanation that maternal activity might be related to a decreased risk of preterm delivery, as the study suggests. Considering the above, some of the premises taken into consideration for this study were the lack of attention to care and delayed follow-up for pregnant women in Romania after several months of lockdown and the disseminated fear of contacting the virus from attending in-person hospital visits among pregnant patients. The current study aims to analyze pregnancies that covered a longer period of the pandemic and determine if singleton live birth outcomes from mothers infected with SARS-CoV-2 differ from mothers that did not contact the virus throughout their pregnancy and if the COVID-19 itself is an independent risk factor for adverse birth outcomes. 

## 2. Materials and Methods

The present population-based cohort study took place in a tertiary hospital from Timisoara, Romania, at the “Dr. Dumitru Popescu” Clinical Hospital of Obstetrics and Gynecology, during a period of five months, between 30 August 2020 and 30 January 2021. The inclusion criteria for pregnant women admitted to our clinic comprised the following: (a) signing informed consent and agreement to participate in the study; (b) giving birth to a single live child; (c) being tested negative or positive for SARS-CoV-2 before admission or at the time of hospital admission, using the standard RT-PCR procedure. A total of 307 pregnant women with COVID-19-associated symptomatology who did not give birth and tested negative for SARS-CoV-2 for at least two consecutive RT-PCR tests were excluded from the study. In addition, a number of 488 pregnant women who gave birth during the study period in our clinic refused to be enrolled in the project, while 24 twin pregnancies were also excluded. At the end of the study period, 938 COVID-19-negative and 101 COVID-19-positive pregnant women met the inclusion criteria.

A dedicated database was created to track SARS-CoV-2 infection during pregnancy. The primary researcher entered data for each pregnancy before and after delivery, with a 6-week follow-up to detect problems or symptomatic infections. We devised an analytical strategy based on the approved empirical techniques [16] and followed established reporting criteria for our findings. We recruited infected pregnant women identified by standard screening for SARS-CoV-2 infection, which was performed on all obstetric patients during pregnancy follow-up until admission to our clinic’s delivery unit between 30 August 2020 and 30 January 2021 (Figure 1). Positive polymerase chain reaction (PCR) results from nasopharyngeal swabs were used to diagnose SARS-CoV-2 infection. The research included all individuals found, regardless of clinical signs and symptoms or the outcome of another serological test. The infection was categorized as COVID-19 with mild symptoms, mild or moderate pneumonia, severe pneumonia, and COVID-19 with septic shock in those individuals having clinical manifestations of SARS-CoV-2 infection, following the WHO guidelines at the time of the study development [17]. A computed tomography (CT) scan was carried for all patients with moderate to severe symptoms, and it did not include the gravid uterus in the field of view.

This prospective study was approved by the Local Committee of Ethics for Scientific Research of “Dr. Dumitru Popescu” Clinical Hospital of Obstetrics and Gynecology operating under provisions of article 167 of the Law number 95 from 2006, art. 28, chapter VIII of order 904/2006 and with EU GCP Directives 2005/28/EC established at the International Conference on Harmonization of Technical Requirements for Registration of Pharmaceuticals for Human Use (ICH), and with the Declaration of Helsinki—Recommendations Guiding Medical Doctors in Biomedical Research Involving Human Subjects. The current study protocol received ethical approval on 10 August 2020, with the approval number of 992.

We compared the clinical and laboratory data of obstetric patients with and without an associated COVID-19 infection using the IBM SPSS v.26 statistical software. The *χ*2 test and Fisher’s exact test were used for categorical variables and Student’s *t*-test or the Mann–Whitney *U*-test for continuous variables. The independent risk factors associated with the COVID-19-related pregnancy outcomes and the associations between maternal COVID-19 status and adverse birth outcomes, adjusted for potential confounding variables, were identified using a multivariate logistic regression model. Risk factors are reported as odds ratios (ORs) with 95% confidence intervals (CIs). The significance threshold was set for α = 0.05. The variables included in the statistical analysis comprised the patients’ age with the following categories: younger than 25 years old (yo), between 25 yo and 34 yo, 35 yo and above; area of residence (urban or rural); level of education (no education, primary education, high school, and higher education—college and above); occupation (student, no occupation, and employed); marital status (single, concubine, and married); gravidity (one, two, three, or more); parity (one, two, three, or more); pregnancy-associated complications (preeclampsia, gestational hypertension, gestational diabetes mellitus, and premature rupture of amniotic membranes); previous cesarean section (yes or no); trimester of COVID-19 diagnosis (1, 2, or 3); type of actual birth (emergency c-section or vaginal delivery); Appearance, Pulse, Grimace, Activity, and Respiration (APGAR) score of the newborn; postpartum maternal complications; and neonatal outcomes (COVID-19 positive newborn, premature birth, intensive care admission, fetal malformations, neonatal sepsis, and neonatal death). Premature birth is generally described, as any birth occurring before 37 weeks of gestation has ended [18]. Preterm prelabor membrane rupture (PROM) is defined as the rupture of the fetal membranes before to the completion of 37 weeks of gestation. This major obstetric issue affects about 3% to 4% of all pregnancies and is directly associated with 40% to 50% of all premature deliveries [19].

## 3. Results

Of the 1039 eligible pregnant women included in the study, 101 (9.7%) were diagnosed with COVID-19 (Figure 1). The general characteristics and demographics of the women enrolled in the study (Table 1) indicated the highest incidence (>70%) of pregnancies eligible for this study in the 24–35-year-old group, without any significant differences between COVID-19-negative and -positive women (*p*-value = 0.152). No differences in distributions for the level of education were observed among our study groups (*p*-value = 0.474), where around 50% of patients had higher education (51% for COVID-19-negative women vs. 47% in COVID-19-positive women), and less than 10% did not have a minimal education (6% in the COVID-19-negative group vs. 9% in the COVID-19-positive group). In addition, 52% of our patients without a confirmed SARS-CoV-2 infection were employed compared to 57% for the positive patients (*p*-value = 0.409). Lastly, regarding the general characteristics of the study groups, there were no statistically significant differences in proportions for the area of residence (*p*-value = 0.195).

The majority of pregnant women enrolled in the study were at the first pregnancy (47% in the COVID-19-negative group vs. 49% in the COVID-19-positive group; *p*-value = 0.792), while also giving birth for the first time (55% vs. 58%; *p*-value = 0.801). Approximately 20% of all patients had three or more gestations, while 10% had three or more live births (Table 2). Regarding the COVID-19-positive patients, 52% were infected in the third trimester, resulting in 2 newborns (2%) being infected with SARS-CoV-2, although without CT chest scan modifications. There were no statistically significant differences observed in the distribution of mothers with a previous c-section, having a rate of 25% in the negative group, compared with 28% in the positive group (*p*-value = 0.408). Comparing obstetrical outcomes for the patients being studied, we encountered significantly more cases necessitating emergency c-sections in the mothers infected with SARS-CoV-2 and significantly more occurrences of PROMs (11% in the COVID-19-positive group vs. 6% in the COVID-19-negative group; *p*-value = 0.049). Postpartum anemia was more frequent after the SARS-CoV-2 infection (*p*-value = 0.004). The newborns of these mothers scored a significantly lower APGAR score than the control group, where 18% were awarded a score of equal or less than 8, versus 11% in the other group (*p*-value = 0.020). Premature births occurred in 15% of cases with COVID-19, rather than just 8% in the live births from COVID-19-negative mothers (*p*-value < 0.001). 

A multivariate logistic regression model (Table 3) was performed using the covariates “type of birth” APGAR score, postpartum anemia, prematurity, and PROM, as previously identified to differ among the two study groups significantly. Except for postpartum anemia, the other variables were statistically significant identified as independently associated with the SARS-CoV-2 infection. Mothers with confirmed COVID-19 had a significantly higher rate of premature birth, either iatrogenic due to associated complications or spontaneously induced. Compared to mothers without COVID-19, those with a confirmed infection had an adjusted OR (AOR) of 1.61 (95% CI: 1.19–2.04) for preterm delivery; an AOR of 2.13 (95% CI: 1.46–2.91) for a lower APGAR score; an AOR of 1.24 (95% CI: 1.09–1.45) for an emergency c-section; and an AOR of 2.46 (95% CI: 2.00–3.19) for PROM. 

## 4. Discussion

### 4.1. Review of the Literature

The existing studies suggest that physiological and immunologic changes associated with pregnancy may increase the risk of pregnant women contracting respiratory viruses [20]. Moreover, initial research on COVID-19 and pregnant women found them to be more vulnerable to infections, have more severe consequences of the disease and have a higher death rate than the general population [21]. The same results were later confirmed by a wide study carried as a multinational cohort comprising 2130 pregnant women [22], describing that COVID-19 diagnosis during pregnancy increased the risk of preeclampsia and eclampsia by 1.76 times, severe infections were 3.38 times more common, and the risk of maternal death due to SARS-CoV-2 infection was 22 times higher when compared to pregnant women without COVID-19 infection. Regarding neonatal outcomes, the same study reported a two-fold higher risk of delivery by c-section, a five-fold increase in neonatal complications, and 1.59 times higher risk of preterm birth. These findings are consistent with our results, although maternal complications were not common or statistically significantly higher than in cases without COVID-19. 

Our study did not report any significant or conclusive findings regarding the vertical transmission of SARS-CoV-2, since there were only two newborns diagnosed with COVID-19. However, other research on this topic [23] determined an approximately 3.2% infection rate by vertical transmission in the third trimester. There, a total of 22 out of 936 neonates were certainly infected with SARS-CoV-2 since they had high IgM titers shortly after birth, while the IgM antibodies are not transferred through the placenta [24]. Therefore, our findings do not go far from this rate of infection, having 2% probable vertical infections. 

Early observations of COVID-19 implications in pregnancy [25] suggested that mothers should abstain from nursing until it is determined that they are no longer infected with COVID-19, and their newborns should be isolated to prevent neonatal transmission. Our study did not include a large sample of patients to make possible the investigation of vertical transmission or breastfeeding transmission of the SARS-CoV-2 virus. However, a later research conducted by Lubbe et al. [26] studied, in a literature review, the effects of breastfeeding on newborns with COVID-19-positive mothers and concluded that SARS-CoV-2 is not spread by breastfeeding, according to the existent findings. The benefits of breastfeeding exceed the hazards associated with the COVID-19 pandemic and may even protect the child and the mother, although general infection control measures should be implemented and carefully followed, such as handwashing before breastfeeding. Breastfeeding should be promoted, mothers and infant dyads should be supported, and skin-to-skin contact should be maintained during the COVID-19 pandemic. The same study results suggest that women who are unable to breastfeed due to illness should be encouraged to collect their milk, and the child should be fed by a healthy person.

### 4.2. Limitations

Possible limitations that our research faces are the chance of abnormal outcomes observed in the pregnant women positive with SARS-CoV-2 infection being caused by the treatment scheme that all COVID-19 patients followed during hospital admission. Thus, the treatment comprising ceftriaxone, fraxiparine, paracetamol, and dexamethasone could be a potential confounding factor for the difference in obstetrical outcomes discovered in the infected patients. Other study limitations include the lack of laboratory data being analyzed, since we only observed the clinical outcomes in relation to the SARS-CoV-2 viral infection. In addition, the study group comprising COVID-19-positive pregnant women was relatively small, as we did not calculate an appropriate sample size but instead included all eligible pregnancies since the rate of COVID-19 infection in the general population and among pregnant women was relatively low.

## 5. Conclusions

This prospective cohort research shows that SARS-CoV-2 infection or diagnosis with COVID-19 during late pregnancy is associated with an increased risk of preterm delivery with a c-section, additionally causing a premature rupture of membranes and determining a lower APGAR score in newborns from mothers with COVID-19.

## Figures and Tables

**Figure 1 medicina-57-00796-f001:**
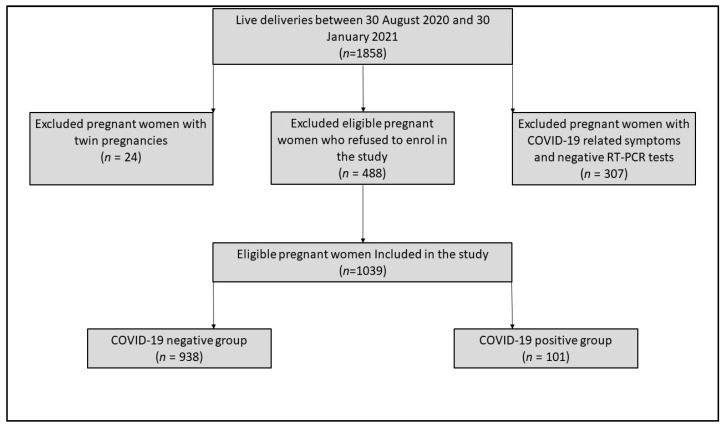
Flow chart of the study participants.

**Table 1 medicina-57-00796-t001:** Maternal general characteristics.

Characteristics	COVID-19-Negative (*n* = 938)	COVID-19-Positive (*n* = 101)	*p*-Value
**Age (years)**			0.152
<25	103 (11%)	8 (8%)	
25–34	703 (75%)	72 (71%)	
>35	132 (14%)	21 (21%)	
**Level of education**			0.474
No education	56 (6%)	9 (9%)	
Primary education	75 (8%)	11 (11%)	
High school	326 (35%)	33 (33%)	
Higher education	478 (51%)	48 (47%)	
**Occupation**			0.409
Student	178 (19%)	20 (20%)	
No occupation	272 (29%)	23 (23%)	
Employed	488 (52%)	58 (57%)	
**Area of residence**			0.195
Urban	591 (63%)	57 (56%)	
Rural	347 (37%)	44 (44%)	

**Table 2 medicina-57-00796-t002:** Obstetrical characteristics and complications.

Characteristics	COVID-19-Negative (*n* = 938)	COVID-19-Positive (*n* = 101)	*p*-Value
**Gravidity**			0.792
One	441 (47%)	50 (49%)	
Two	310 (33%)	30 (30%)	
Three or more	187 (20%)	21 (21%)	
**Parity**			0.801
One	516 (55%)	59 (58%)	
Two	328 (35%)	33 (33%)	
Three or more	94 (10%)	9 (9%)	
**Previous cesarean section**			0.408
Yes	234 (25%)	29 (28%)	
No	704 (75%)	72 (72%)	
**Trimester of COVID-19 diagnosis**			
1	-	27 (26%)	
2	-	22 (22%)	
3	-	52 (52%)	
**Type of birth**			0.001
Vaginal delivery	779 (83%)	71 (70%)	
Emergency c-section	159 (17%)	30 (30%)	
**APGAR score**			0.020
≥9	835 (89%)	83 (83%)	
7–8	66 (7%)	8 (8%)	
≤6	37 (4%)	10 (10%)	
**Postpartum maternal complications**			
Anemia	263 (28%)	42 (42%)	0.004
Infection	413 (44%)	51 (51%)	0.214
Perineal laceration	366 (39%)	30 (30%)	0.066
**Neonatal outcomes**			
SARS-CoV-2 infection	-	2 (2%)	
Prematurity	75 (8%)	15 (15%)	<0.001
ICU admission	27 (3%)	6 (6%)	0.095
Fetal malformations	19 (2%)	2 (2%)	0.975
Neonatal sepsis	56 (6%)	5 (5%)	0.678
Neonatal death	4 (0.5%)	1 (1%)	0.436
**Pregnancy-associated complications**			
Gestational hypertension	28 (3%)	6 (6%)	0.112
Preeclampsia	19 (2%)	2 (2%)	0.975
Gestational diabetes mellitus	66 (7%)	5 (5%)	0.429
PROM	55 (6%)	11 (11%)	0.049

PROM = premature rupture of membranes; ICU = intensive care unit.

**Table 3 medicina-57-00796-t003:** Multivariate analysis.

	COVID-19-Negative	COVID-19-Positive
**Preterm birth**	75 (8%)	15 (15%)
OR (95% CI)	1.00	1.43 (1.08–1.79)
Adjusted OR (95% CI)	1.00	1.61 (1.19–2.04)
**APGAR score of <9**	103 (11%)	18 (18%)
OR (95% CI)	1.00	1.93 (1.21–2.40)
Adjusted OR (95% CI)	1.00	2.13 (1.46–2.91)
**Emergency c-section**	159 (17%)	30 (30%)
OR (95% CI)	1.00	1.12 (1.04–1.37)
Adjusted OR (95% CI)	1.00	1.24 (1.09–1.45)
**PROM**	55 (6%)	11 (11%)
OR (95% CI)	1.00	2.28 (1.65–3.01)
Adjusted OR (95% CI)	1.00	2.46 (2.00–3.19)

## Data Availability

The data presented in this study are available on request from the corresponding author.

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
