# Peer review of "Exploring Pregnancy Outcomes Associated with SARS-CoV-2 Infection"

_medicina, 2021, doi:10.3390/medicina57080796_

Round 1

Reviewer 1 Report

an interesting paper.

please give more details regarding -preterm labor and PPROM

weeks of gestation for these pregnancies? 

did these women have risk factors fo preterm labor and PPROM ?

did they have other causes for preterm labor and PPROM? for example :abruptio placenta? chorioamnionitis? if so 

whether the placenta was sent for histological examination? cultures?

Author Response

Dear reviewer,

Thank you for considering our manuscript, as we appreciate your efforts to analyse and improve our paper. Thus, based on the feedback received during the review phase, our team had carefully revised the article with the following changes:

Thank you for recommending the literature to describe the impact that COVID-19 had on infertile women in terms of risk for infertility and on IVF management. However, we consider this to be out of topic, since we evaluate only pregnancies that concluded in live births.

The abstract section was slightly modified.

Line 58: added “and”

Line 81: added “decreased risk of preterm delivery”

Line 82: We have removed reference number 16

Line 84: We have added on the premises of developing the current study considering the impact of the COVID-19 pandemic on pregnant women in Romania.

Line 105: added “At the end of the study period, 938 COVID-19-negative and 101 COVID-19-positive pregnant women met the inclusion criteria.”

Line 151: premature birth was defined according to Quinn JA, Munoz FM, Gonik B, et al. Preterm birth: Case definition & guidelines for data collection, analysis, and presentation of immunisation safety data. Vaccine. 2016;34(49):6047-6056. doi:10.1016/j.vaccine.2016.03.045

Line 152: PROM was defined according to Menon R, Richardson LS. Preterm prelabor rupture of the membranes: A disease of the fetal membranes. Semin Perinatol. 2017;41(7):409-419. doi:10.1053/j.semperi.2017.07.012

We did not include the weeks of gestation for these patients, although they were grouped by trimester of COVID-19 diagnosis, which should suffice.

The risk factors were not fully assessed, but we counted the cases of gestational hypertension, preeclampsia, gestational diabetes and PROM as potential risk factors.

We did not take into consideration whether the placenta was sent for histological examination.

Best regards,

The authors

Reviewer 2 Report

Authors presented a cohort study on maternal and fetal outcomes associated with SARS-COV-2 infection. The study is not very novel since after one and almost a half year we do already know the huge impact the infection had on the management of pregnant women infected and the increased risk for pregnancy and neonatal complications. However, the work is quite well written and the study well conducted. Here I have some suggestions and remarks.

In the introduction authors could focus the attention on their national covid-19 situation, both in general population and in pregnant women, to justify the need to perform their study and later disclose their results, given the not-novel findings. I acknowledge that one of the reasons was the uncertainty in some complication like the preterm delivery, but I think that in this way they would not pretend to give an answer to the issue, but only add their point of view to an unsolved problem (the impact of COVID-19 on preterm delivery).

Moreover, initially you could also describe the impact that COVID-19 had on infertile women both in terms of risk for infertility and on IVF management

  1. Khalili MA, Leisegang K, Majzoub A, Finelli R, Panner Selvam MK, Henkel R, Mojgan M, Agarwal A. Male Fertility and the COVID-19 Pandemic: Systematic Review of the Literature. World J Mens Health. 2020 Oct;38(4):506-520. doi: 10.5534/wjmh.200134. Epub 2020 Aug 14. PMID: 32814369; PMCID: PMC7502312.
  2. Carbone L, Conforti A, LA Marca A, Cariati F, Vallone R, Raffone A, Buonfantino C, Palese M, Mascia M, DI Girolamo R, Capuzzo M, Esteves SC, Alviggi C. The negative impact of most relevant infections on fertility and Assisted Reproduction Technology. Minerva Obstet Gynecol. 2021 Jun 17. doi: 10.23736/S2724-606X.21.04870-3. Epub ahead of print. PMID: 34137567.
  3. Alviggi C, Esteves SC, Orvieto R, Conforti A, La Marca A, Fischer R, Andersen CY, Bühler K, Sunkara SK, Polyzos NP, Strina I, Carbone L, Bento FC, Galliano D, Yarali H, Vuong LN, Grynberg M, Drakopoulos P, Xavier P, Llacer J, Neuspiller F, Horton M, Roque M, Papanikolaou E, Banker M, Dahan MH, Foong S, Tournaye H, Blockeel C, Vaiarelli A, Humaidan P, Ubaldi FM; POSEIDON (Patient-Oriented Strategies Encompassing IndividualizeD Oocyte Number) group. COVID-19 and assisted reproductive technology services: repercussions for patients and proposal for individualized clinical management. Reprod Biol Endocrinol. 2020 May 13;18(1):45. doi: 10.1186/s12958-020-00605-z. PMID: 32404170; PMCID: PMC7218705.
  4. Vaiarelli A, Bulletti C, Cimadomo D, Borini A, Alviggi C, Ajossa S, Anserini P, Gennarelli G, Guido M, Levi-Setti PE, Palagiano A, Palermo R, Savasi V, Pellicer A, Rienzi L, Ubaldi FM. COVID-19 and ART: the view of the Italian Society of Fertility and Sterility and Reproductive Medicine. Reprod Biomed Online. 2020 Jun;40(6):755-759. doi: 10.1016/j.rbmo.2020.04.003. Epub 2020 Apr 8. PMID: 32354663; PMCID: PMC7141636.

Among the aim of the study, again, I think it would be preferable to stress the concept of analyzing the impact in your nation, to see the comparison with other reports concerning maternal outcomes of COVID-19.

Regarding the Discussion section:

First of all, I suggest to divide the discussion into the following paragraphs:

  • Main findings, in which authors should repeat the most important results they observed in their cohort, both if statistically significant or not
  • Strength and limitations, in which authors acknowledge why their study is good (rigorous study methods, first study in their country, etc.) and limitations (small sample size and others)
  • Implications and future directions (here authors can describe what was already found by others and what literature show on the topic, suggesting something in relation to tracking new cases, vaccination during pregnancy, other studies with larger samples, etc.)

In relation to vaccination the main problem is the low acceptance rate in pregnant women

(please consider these studies)

  • Mohan S, Reagu S, Lindow S, Alabdulla M. COVID-19 vaccine hesitancy in perinatal women: a cross sectional survey. J Perinat Med. 2021 Apr 27. doi: 10.1515/jpm-2021-0069. Epub ahead of print. PMID: 33905622.
  • Goncu Ayhan S, Oluklu D, Atalay A, Menekse Beser D, Tanacan A, Moraloglu Tekin O, Sahin D. COVID-19 vaccine acceptance in pregnant women. Int J Gynaecol Obstet. 2021 Apr 19. doi: 10.1002/ijgo.13713. Epub ahead of print. PMID: 33872386.
  • Mappa I, Luviso M, Distefano FA, Carbone L, Maruotti GM, Rizzo G. Women perception of SARS-CoV-2 vaccination during pregnancy and subsequent maternal anxiety: a prospective observational study. J Matern Fetal Neonatal Med. 2021 Apr 11:1-4. doi: 10.1080/14767058.2021.1910672. Epub ahead of print. PMID: 33843419.
  • Carbone L, Mappa I, Sirico A, Girolamo RD, Saccone G, Mascio DD, Donadono V, Cuomo L, Gabrielli O, Migliorini S, Luviso M, D'antonio F, Rizzo G, Maruotti GM. Pregnant women perspectives on SARS-COV-2 vaccine: Condensation: Most of Italian pregnant women would not agree to get the SARS-COV-2 vaccine, irrespective of having features of high risk themselves, or being high-risk pregnancies. Am J Obstet Gynecol MFM. 2021 Mar 23:100352. doi: 10.1016/j.ajogmf.2021.100352. Epub ahead of print. PMID: 33771762; PMCID: PMC7985679

In addition, since in the methods you admit the use of CT scan in pregnant women, in the “future directions paragraph” you could mention that CT scan can be applied during pregnancy but there are unpredictable although low risk due to radiations, which could be handled also with the use of lung ultrasound, with a useful application also in the triage of symptomatic women. Please consider

  1. Yassa M, Yirmibes C, Cavusoglu G, Eksi H, Dogu C, Usta C, Mutlu M, Birol P, Gulumser C, Tug N. Outcomes of universal SARS-CoV-2 testing program in pregnant women admitted to hospital and the adjuvant role of lung ultrasound in screening: a prospective cohort study. J Matern Fetal Neonatal Med. 2020 Nov;33(22):3820-3826. doi: 10.1080/14767058.2020.1798398. Epub 2020 Jul 28. PMID: 32691641.
  2. Buonsenso D, Raffaelli F, Tamburrini E, Biasucci DG, Salvi S, Smargiassi A, Inchingolo R, Scambia G, Lanzone A, Testa AC, Moro F. Clinical role of lung ultrasound for diagnosis and monitoring of COVID-19 pneumonia in pregnant women. Ultrasound Obstet Gynecol. 2020 Jul;56(1):106-109. doi: 10.1002/uog.22055. PMID: 32337795; PMCID: PMC7267364.
  3. Deng Q, Cao S, Wang H, Zhang Y, Chen L, Yang Z, Peng Z, Zhou Q. Application of quantitative lung ultrasound instead of CT for monitoring COVID-19 pneumonia in pregnant women: a single-center retrospective study. BMC Pregnancy Childbirth. 2021 Mar 26;21(1):259. doi: 10.1186/s12884-021-03728-2. PMID: 33771120; PMCID: PMC7997654.
  4. Carbone L, Esposito R, Raffone A, Verrazzo P, Carbone IF, Saccone G. Proposal for radiologic diagnosis and follow-up of COVID-19 in pregnant women. J Matern Fetal Neonatal Med. 2020 Jul 16:1-2. doi: 10.1080/14767058.2020.1793325. Epub ahead of print. PMID: 32669006.

Furthermore, I have some grammatical and syntactical remarks as follows:

Abstract section, line 12: “more than one year” in place of “almost”

Line 19: “center” and non “centric”

Line 24-25: “significant” and not “significantly”

Lines 24-29: please modify this section, since you repeat twice the increased rate of preterm birth, low apgar score, premature rupture of membranes and c-section. There is no point to repeat it twice in a 5-lines abstract results section. Explain the results just once.

Introduction, line 58: I think you missed an “and” between “deliveries other”

Line 78, when authors say “increased risk of decreased preterm delivery”, I imagine authors meant a decreased risk of preterm delivery. Please rewrite and clarify.

Line 79: I do not agree when authors say “Systematic reviews [16] did not clarify if COVID-19 can complicate pregnancy”; first of all, the reference n. 16 is not a systematic review on the topic COVID-19 and pregnancy outcomes but only a guideline on reporting observational studies. Moreover, I am sure authors do acknowledge that acquiring COVID-19 infection during pregnancy increases the overall risk of adverse pregnancy outcomes, independently from a specific outcome or condition. If authors refer to the preterm birth rate, which they already discussed in the above lines saying that different studies had different results, they should specify that; in this regard, however, I think they already explained the uncertainty of literature and therefore there is no need to repeat it. If instead this is not the meaning authors wanted to give, please rewrite and explain.

Please have a look also at :

1) Khalil A, Kalafat E, Benlioglu C, O'Brien P, Morris E, Draycott T, Thangaratinam S, Le Doare K, Heath P, Ladhani S, von Dadelszen P, Magee LA. SARS-CoV-2 infection in pregnancy: A systematic review and meta-analysis of clinical features and pregnancy outcomes. EClinicalMedicine. 2020 Aug;25:100446. doi: 10.1016/j.eclinm.2020.100446. Epub 2020 Jul 3. PMID: 32838230; PMCID: PMC7334039.

2) Lassi ZS, Ana A, Das JK, Salam RA, Padhani ZA, Irfan O, Bhutta ZA. A systematic review and meta-analysis of data on pregnant women with confirmed COVID-19: Clinical presentation, and pregnancy and perinatal outcomes based on COVID-19 severity. J Glob Health. 2021 Jun 30;11:05018. doi: 10.7189/jogh.11.05018. PMID: 34221361; PMCID: PMC8248750.

3) Allotey J, Stallings E, Bonet M, Yap M, Chatterjee S, Kew T, Debenham L, Llavall AC, Dixit A, Zhou D, Balaji R, Lee SI, Qiu X, Yuan M, Coomar D, Sheikh J, Lawson H, Ansari K, van Wely M, van Leeuwen E, Kostova E, Kunst H, Khalil A, Tiberi S, Brizuela V, Broutet N, Kara E, Kim CR, Thorson A, Oladapo OT, Mofenson L, Zamora J, Thangaratinam S; for PregCOV-19 Living Systematic Review Consortium. Clinical manifestations, risk factors, and maternal and perinatal outcomes of coronavirus disease 2019 in pregnancy: living systematic review and meta-analysis. BMJ. 2020 Sep 1;370:m3320. doi: 10.1136/bmj.m3320. PMID: 32873575; PMCID: PMC7459193.

Materials and methods section, lines 98-99; I would modify as follows: “At the end of the study period, 938 COVID-19-negative and 101 COVID-19-positive pregnant women met the inclusion criteria.”

Line 138: “gestational hypertension” in place of “gestational high blood pressure”

Line 139: “cesarean” and not “cesarian”

Results section, line 149: “years old” in place of “age”

Author Response

(The authors gave the same response as above.)
